# Learning Contrastive Embedding
# in Low-Dimensional Space

**Shuo Chen**[†], **Chen Gong**[§], **Jun Li**[§], **Jian Yang**[§], **Gang Niu**[†], **Masashi Sugiyama**[‡]

## Abstract

*Contrastive learning* (CL) pretrains feature embeddings to *scatter* instances in the feature space so that the training data can be well discriminated. Most existing CL techniques usually encourage learning such feature embeddings in the *high-dimensional space* to maximize the instance discrimination. However, this practice may lead to undesired results where the scattering instances are *sparsely distributed* in the high-dimensional feature space, making it difficult to capture the underlying similarity between pairwise instances. To this end, we propose a novel framework called *contrastive learning with low-dimensional reconstruction* (CLLR), which adopts a regularized projection layer to *reduce* the dimensionality of the feature embedding. In CLLR, we build the *sparse/low-rank* regularizer to adaptively reconstruct a low-dimensional projection space while preserving the basic objective for instance discrimination, and thus successfully learning contrastive embeddings that alleviate the above issue. Theoretically, we prove a tighter error bound for CLLR; empirically, the superiority of CLLR is demonstrated across multiple domains. Both theoretical and experimental results emphasize the significance of learning low-dimensional contrastive embeddings.

## 1  Introduction

Recently, unsupervised learning approaches have been greatly promoted by the *contrastive learning* (CL), which shows encouraging performance compared to fully supervised approaches [8, 21]. CL pretrains deep neural networks with unlabeled instances, and the learned feature embeddings can be directly used to extract features from the raw data [35]. Thereby, CL has been successfully applied in many downstream recognition tasks such as classification [42], retrieval [41], and clustering [3].

As an unsupervised learning problem setting where the human annotation is not available, CL approaches usually consider building the *pseudo supervision* in their learning objectives [36, 19], and thus CL is also regarded as a *self-supervised learning* approach. Originally, the pseudo supervision of CL is to push away each pair of instances to scatter data points in the feature space, by which all instances in the training data can be well discriminated (*i.e.*, the *instance discrimination*) [14, 40]. This original design has been empirically validated to be particularly effective in the representation learning [28, 6], and has also been theoretically proved to approximate an *unbiased* supervised learning objective [32, 11]. Many recent efforts have increasingly focused on two different directions to further improve the performance of CL. The first one is to introduce plentiful data-augmentation

---

[†]S. Chen and G. Niu are with RIKEN Center for Advanced Intelligence Project (AIP), Japan (E-mail: {shuo.chen.ya@riken.jp, gang.niu.ml@gmail.com}).

[§]C. Gong, J. Li, and J. Yang are with the PCA Lab, Key Lab of Intelligent Perception and Systems for High-Dimensional Information of Ministry of Education, and Jiangsu Key Lab of Image and Video Understanding for Social Security, School of Computer Science and Engineering, Nanjing University of Science and Technology, China (E-mail: {junli, chen.gong, csjyang}@njust.edu.cn).

[‡]M. Sugiyama is with RIKEN Center for Advanced Intelligence Project (AIP), Japan; and also with the Graduate School of Frontier Sciences, The University of Tokyo, Japan (E-mail: sugi@k.u-tokyo.ac.jp).

36th Conference on Neural Information Processing Systems (NeurIPS 2022).

to generate the *positive pair* which consists of each instance and its perturbation [35, 28]. Then, any two instances in the training data are regarded as the *negative pair*, and the objective of *metric learning* [33, 43] can be used to learn feature embeddings that distinguish positive pairs and negative pairs. Nevertheless, the negative pairs built in CL are inherently noisy because they contain false negatives consisted of semantically similar instances [30]. Therefore, the second way to improve the performance of CL is to reduce the impact of false negative pairs. To this end, some recent works convert it to *positive-unlabeled learning* [11, 27] and clustering problems [3, 46] to reweight the importance of negative pairs, and thus constraining the undesired repelling of negative pairs [3, 46].

Although the existing methods have achieved promising results to some extent, their reliabilities highly depend on the effectiveness of instance discrimination [20, 32]. However, recent works usually encourage learning contrastive embedding in *high-dimensional space* to maximally discriminate instances, so that the dimensionality of self-supervised contrastive embedding [7, 9] is set to be much larger than the dimensionality of traditional fully supervised embedding [10, 44]. This practice makes data points *sparsely distribute* in the feature space (which is similar to the *curse of dimensionality* [18]), and thereby the corresponding CL methods may fail to capture the intrinsic similarity between pairwise instances. Such a problem can hardly be solved by simply setting a low dimensionality for the output layer, as it will cause the *dimensional collapse* with insufficient instance discrimination [20]. Some popular compression approaches such as distillation techniques [5, 47] enable us to train small networks under the supervision of original contrastive embeddings, yet the improper similarity predictions can still be inherited from the original networks. Therefore, a new CL method is desired to effectively learn the low-dimensional feature embedding.

In this paper, we propose a novel framework dubbed *contrastive learning with low-dimensional reconstruction* (CLLR) to explicitly address the above issue caused by the high dimensionality in CL. Specifically, we introduce a new sparse projection layer to reconstruct the features of instances in low-dimensional space and meanwhile scatter all instances in the original high-dimensional space (see Fig. 1). Then, we obtain the low-dimensional contrastive embedding which can also effectively distinguish instances in the training data. Theoretically, we prove a lower bound for the min-max distance ratio of the learned contrastive embedding, which ensures that CLLR can better capture the instance similarity than the existing CL models.

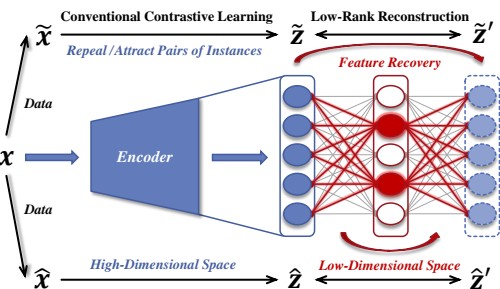

Figure 1: Conceptual illustration of our proposed CLLR. In our method, we discriminate all instances in high-dimensional space and introduce a sparse projection layer (the red part) to reconstruct the features of instances in the low-dimensional latent space.

Experimentally, our approach consistently improves the state-of-the-art methods on vision, language, and reinforcement learning benchmarks. To the best of our knowledge, we are the first to propose learning the original contrastive embedding in low-dimensional space. The proposed method is very generic, so it can be applied in many existing CL models. Our main contributions are summarized as: **I).** we propose a novel framework to enhance the generalization ability of contrastive learning via introducing a sparse / low-rank regularized projection layer to adaptively reduce the high dimensionality of contrastive embedding; **II).** we establish complete theoretical guarantees for our method by analyzing the error bound of distance predictions and the convergence of the learning algorithm, respectively; **III).** we conduct extensive experiments on real-world datasets to validate the superiority of our method over the state-of-the-art CL approaches, and the results consistently emphasize the necessity / significance of learning low-dimensional contrastive embeddings.

**Notations.** We write matrices and vectors as bold uppercase characters and bold lowercase characters, respectively. We denote the training dataset $\mathcal{X} = \{\boldsymbol{x}_i \in \mathbb{R}^m | i = 1, 2, \ldots, N\}$ where $m$ is the data dimensionality and $N$ is the sample size. Operators $\|\cdot\|_2$, $\|\cdot\|_{2,1}$, and $\|\cdot\|_*$ denote the vector/matrix $\ell_2$-norm, $\ell_{2,1}$-norm (*i.e.*, the sum of $\ell_2$-norm for columns), and nuclear-norm, respectively.

## 1.1 Background & Related Work

In this subsection, we briefly review the background of contrastive learning and the related work.

**Instance Discrimination & Contrastive Learning.** As a popular unsupervised / self-supervised learning approach, the basic goal of contrastive learning (CL) algorithm is to learn a generic feature embedding $\boldsymbol{\Phi} \colon \mathbb{R}^m \mapsto \mathbb{R}^H$, which transforms the data point from $m$-dimensional sample space to $H$-dimensional embedding space. The primitive CL method called instance discrimination learns such an embedding by directly enlarging the following distance between each pair of two instances $\boldsymbol{x}_i$ and $\boldsymbol{x}_j$ in the training data [14, 40]

$$\mathcal{D}_{\boldsymbol{\Phi}}(\boldsymbol{x}_i, \boldsymbol{x}_j) = \|\boldsymbol{\Phi}(\boldsymbol{x}_i) - \boldsymbol{\Phi}(\boldsymbol{x}_j)\|_2 / H, \tag{1}$$

where $H$ is the dimensionality of the learned feature embedding. The design philosophy for instance discrimination is that when we scatter all instances in the feature space, the characteristic of each instance are captured and thus the training data can be well memorized by the neural network [20]. When we further generate the positive pairs $(\boldsymbol{x}, \boldsymbol{x}^+)$ by combining each single instance $\boldsymbol{x}$ and its perturbation $\boldsymbol{x}^+$, we are able to use the *noise contrastive estimation* (NCE) loss [16] to learn a feature embedding $\boldsymbol{\Phi}$ from positive and negative pairs. In this paper, we focus on such a NCE loss which has the form of $\mathcal{L}_{\mathrm{NCE}}(\boldsymbol{\Phi}) = \mathbb{E}_{\boldsymbol{x}, \boldsymbol{x}_j^- \in \mathcal{X}}[-\log(\mathrm{e}^{\boldsymbol{\Phi}(\boldsymbol{x})^\top \boldsymbol{\Phi}(\boldsymbol{x}^+)} / (\mathrm{e}^{\boldsymbol{\Phi}(\boldsymbol{x})^\top \boldsymbol{\Phi}(\boldsymbol{x}^+)} + \sum_{j=1}^n \mathrm{e}^{\boldsymbol{\Phi}(\boldsymbol{x})^\top \boldsymbol{\Phi}(\boldsymbol{x}_j^-)}))]$. Here instances $\boldsymbol{x}$ and $\{\boldsymbol{x}_j^-\}_{j=1}^n$ are uniformly sampled from the training data $\mathcal{X}$, and $n$ is the batch size.

Admittedly, as the original prototype of CL, instance discrimination is very critical to ensuring the effectiveness of most CL methods. However, the feature dimensionality settings in existing CL methods are usually very high (*e.g.*, 2048-dimension and 4096-dimension in [7, 9] ), which are much larger than the feature dimensionality in most fully supervised learning methods (*e.g.*, 512-dimension and 1024-dimension in [10, 15]). We demonstrate that learning contrastive embeddings in such high-dimensional space can be weak in capturing the similarity between pairwise instances. To address this issue, in this paper, we propose a novel framework to learn contrastive embedding in low-dimensional space, which uses a *sparse / low-rank* regularized projection layer for reconstruction.

**PCA & Autoencoder.** As a classical unsupervised / self-supervised learning method, *principal component analysis* (PCA) has shown promising results in many machine learning tasks [39, 45, 4]. Actually, PCA shares very similar motivation with the instance discrimination of CL. It is well-known that PCA seeks for a vector $\boldsymbol{p} \in \mathbb{R}^m$ to scatter instances in the projection space by maximizing the variance $\mathbb{E}_{\boldsymbol{x} \in \mathcal{X}}[(\boldsymbol{x} - \overline{\boldsymbol{x}})^\top \boldsymbol{p} \boldsymbol{p}^\top (\boldsymbol{x} - \overline{\boldsymbol{x}})]$, where $\overline{\boldsymbol{x}} \in \mathbb{R}^m$ is the mean of all instances in the training data $\mathcal{X}$. Enlarging such a variance is quite similar to the instance discrimination of CL which also pushes away data pairs to scatter instances. PCA has another reconstruction based form $\mathbb{E}_{\boldsymbol{x} \in \mathcal{X}}[\|\boldsymbol{P}\boldsymbol{P}^\top \boldsymbol{x} - \boldsymbol{x}\|_2^2]$ which is equivalent to the variance maximization ($\boldsymbol{P} \in \mathbb{R}^{m \times l}$ is the projection matrix and $l \in \mathbb{Z}_+$ is the dimensionality of orthogonal space). To further improve the fitting ability of PCA for complex data, the non-linear extension Autoencoder introduces the non-linear activation function $\sigma$ and two different projection matrices $\boldsymbol{P}$ and $\boldsymbol{P}'$ to reconstruct the training data in the by minimizing the objective $\mathbb{E}_{\boldsymbol{x} \in \mathcal{X}}[\|\sigma(\boldsymbol{P}'^\top \sigma(\boldsymbol{P}^\top \boldsymbol{x})) - \boldsymbol{x}\|_2^2]$. Some further extensions such as *masked autoencoder* (MAE) [17] achieved very promising results in several downstream tasks.

In this paper, we are inspired by PCA / Autoencoder to reduce the dimensionality of contrastive embedding based on a sparse / low-rank regularized reconstruction loss. Interestingly, from this perspective, our method can also be regarded as a *natural combination* of two main existing self-supervised learning approaches.

## 2 Methodology

In this section, we first investigate the distribution of instances scattered by CL in the high-dimensional feature space. After that, we propose a novel framework dubbed contrastive learning with low-dimensional reconstruction by introducing a new sparse projection layer. The learning objective and the corresponding optimization algorithm are finally designed with convergence guarantee.

### 2.1 Motivation

As we mentioned before, the contrastive embedding $\boldsymbol{\Phi}$ maps an $m$-dimensional instance into the $H$-dimensional feature space. Now we want to investigate the distribution of data points in such an $H$-dimensional space. We consider the $H$-dimensional *hypercube* and its *inscribed-suprasphere*. We suppose that the edge length of the $H$-dimensional hypercube is $2r$, and the radius of its inscribed-

suprasphere will be $r$. Then their corresponding volumes in the high-dimensional space are

$$\mathcal{V}_{\text{cube}}(H) = (2r)^H \quad \text{and} \quad \mathcal{V}_{\text{sphere}}(H) = (2r^H \pi^{H/2})/(H \cdot \Gamma(H/2)), \tag{2}$$

respectively, where $\Gamma(\cdot)$ is the gamma function [37] having a form of $\Gamma(z) = \int_0^\infty t^{z-1}\mathrm{e}^{-t}dt$. We further study the ratio of the suprasphere volume to the hypercube volume. We let $H \to \infty$ and the formulation $\lim_{H\to\infty} \mathcal{V}_{\text{sphere}}(H)/\mathcal{V}_{\text{cube}}(H)$ equals to

$$\lim_{H\to\infty} (\pi^{H/2}/(H \cdot \Gamma(H/2)))/2^{H-1} \leq \lim_{H\to\infty} \pi^{(H-1)/2}/2^{H-1} = \lim_{H\to\infty} (\pi/4)^{(H-1)/2} = 0, \tag{3}$$

and thus we have $\lim_{H\to\infty} \mathcal{V}_{\text{sphere}}(H)/\mathcal{V}_{\text{cube}}(H) = 0$ by using the fact that $\mathcal{V}_{\text{sphere}}(H)/\mathcal{V}_{\text{cube}}(H) \geq 0$. This result of volume ratio clearly reveals that the proportion of the inscribed-suprasphere in the hypercube will gradually converge to $0$ with the increase of the dimensionality $H$. It means that, in the high-dimensional hypercube, a random given data point is less likely to appear *inside* of the inscribed-suprasphere (*i.e.*, in the *central area* of the hypercube) but it will usually exist *outside* of the inscribed-suprasphere (*i.e.*, in the *corner area* of the hypercube).

However, the learning objective of CL expects to scatter all instances in the $H$-dimensional hypercube, and thus making the $N$ instances *sparsely distribute* in the $\widehat{N} = 2^H$ corners. Specifically, for the common dimensionality setting $H = 2048$ in popular CL methods, we have that

$$\widehat{N} = 2^H = 2^{2048} = 16^{512} \gg 10^{512} \gg 10^6 = N, \tag{4}$$

which implies that the corner number $\widehat{N}$ is *significantly larger* than the sample number $N$. In this case, the distribution of instances in the feature space will be very sparse, and all instances are far away from each other. Thereby the learning algorithm can hardly capture the intrinsic similarity between intra-class instances, and the downstream recognition tasks will be affected.

To be more religious, we consider the *min-max distance ratio* to investigate the *distance contrast* in the high-dimensional space. For the *independent and identically distributed* (i.i.d.) instances $\boldsymbol{x}, \boldsymbol{x}_i \in \mathbb{R}^m$ $(i=1,2,\ldots,n)$, their embeddings $\boldsymbol{\Phi}(\boldsymbol{x})$ and $\boldsymbol{\Phi}(\boldsymbol{x}_i)$ are also i.i.d. no matter how the embedding is learned [13]. The following Theorem 1 reveals that the minimal distance $\mathcal{D}_{\boldsymbol{\Phi}}^{\min}(H)$ and the maximal distance $\mathcal{D}_{\boldsymbol{\Phi}}^{\max}(H)$ tend to be equivalent in the high-dimensional space [1] (provided that $\boldsymbol{\Phi}(\boldsymbol{x})$ and $\boldsymbol{\Phi}(\boldsymbol{x}_i)$ are

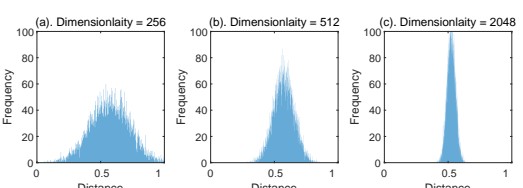

Figure 2: Distance distributions of contrastive embeddings learned on *STL-10* with different feature dimensionalities 256, 512, and 2048.

i.i.d. to each other), so the similarity between pairwise instances can hardly provide contrast to discriminate the intra-class and inter-class (as shown by the distance distributions in Fig. 2).

**Theorem 1.** *For any given i.i.d. random data points $\boldsymbol{x}, \boldsymbol{x}_1, \boldsymbol{x}_2, \ldots, \boldsymbol{x}_n \in \mathbb{R}^m$, we denote $\mathcal{D}_{\boldsymbol{\Phi}}^{\max}(H) = \max\{\mathcal{D}_{\boldsymbol{\Phi}}(\boldsymbol{x},\boldsymbol{x}_i)|i=1,\ldots,n\}$ and $\mathcal{D}_{\boldsymbol{\Phi}}^{\min}(H) = \min\{\mathcal{D}_{\boldsymbol{\Phi}}(\boldsymbol{x},\boldsymbol{x}_i)|i=1,\ldots,n\}$. Then we have that $\lim_{H\to\infty}\{\mathrm{var}[\mathcal{D}_{\boldsymbol{\Phi}}(\boldsymbol{x},\boldsymbol{x}_i)/\mathbb{E}(\mathcal{D}_{\boldsymbol{\Phi}}(\boldsymbol{x},\boldsymbol{x}_i))]\} = 0$ and*

$$\mathcal{P}\left\{\lim_{H\to\infty}(\mathcal{D}_{\boldsymbol{\Phi}}^{\max}(H) - \mathcal{D}_{\boldsymbol{\Phi}}^{\min}(H))/\mathcal{D}_{\boldsymbol{\Phi}}^{\min}(H) = 0\right\} = 1, \tag{5}$$

*where the distance function $\mathcal{D}_{\boldsymbol{\Phi}}(\cdot,\cdot)$ is defined in Eq. (1) and the feature embedding $\boldsymbol{\Phi}$ is learned from the training data and independent to the data points $\boldsymbol{x}, \boldsymbol{x}_1, \boldsymbol{x}_2, \ldots, \boldsymbol{x}_n$.*

In summary, from the above analytical results, we can clearly find that it is very necessary to constrain the dimensionality of existing CL approaches in a reasonable range. Motivated by this, in the next subsection, we provide the formulation of our proposed framework CLLR which reduces the dimensionality of contrastive embedding by a sparse projection layer.

## 2.2  Formulation

As we discussed in the previous subsection, the feature embedding $\boldsymbol{\Phi}$ transforms the raw data from $m$-dimensional space into $H$-dimensional space, where $\boldsymbol{\Phi}$ is learned by the NCE loss. To avoid high-dimensional features, we may directly reduce the dimensionlaity of the output layer, but this will cause the dimensional collapse with insufficient instance discrimination (as we discussed in

Section 1). Thereby, we consider to use an additional matrix $\boldsymbol{L} \in \mathbb{R}^{H \times H}$ to transform the feature embedding result $\boldsymbol{\Phi}(\boldsymbol{x})$ into the latent vector $\boldsymbol{L} \cdot \boldsymbol{\Phi}(\boldsymbol{x})$, and then we minimize $\|\boldsymbol{L}^\top \boldsymbol{L} \cdot \boldsymbol{\Phi}(\boldsymbol{x}) - \boldsymbol{\Phi}(\boldsymbol{x})\|_2^2$, encouraging the latent vector to preserve the useful information in $\boldsymbol{\Phi}(\boldsymbol{x})$. When we further introduce the low-rank constraint for $\boldsymbol{L}$, we can obtain a low-dimensional latent space for contrastive learning.

$\ell_{2,1}$-**Norm based Regularization.** The row (column) sparsity is a long-standing concept which aims to maintain very few non-zero columns for a matrix. When we employ the well-known $\ell_{2,1}$-norm to restrict the projection matrix $\boldsymbol{L}$, we can certainly have a column-sparse $\boldsymbol{L}$ which selects the important features in $\boldsymbol{\Phi}(\boldsymbol{x}) \in \mathbb{R}^H$ corresponding to the non-zero columns. Then we use the selected features to reconstruct the original feature embedding, *i.e.*,

$$\mathcal{R}_{2,1}(\boldsymbol{\Phi}, \boldsymbol{L}) = \mathbb{E}_{\boldsymbol{x} \in \mathcal{X}}[\,\|\boldsymbol{L}^\top \boldsymbol{L} \cdot \boldsymbol{\Phi}(\boldsymbol{x}) - \boldsymbol{\Phi}(\boldsymbol{x})\|_2^2\,] + \alpha \|\boldsymbol{L}\|_{2,1}, \tag{6}$$

where $\boldsymbol{L} \in \mathbb{R}^{H \times H}$ and $\alpha > 0$ is tuned by users. Note that the column sparsity is just a special case of the low-rank, but considering its good usability, here we can easily obtain a low-dimensional feature embedding $\boldsymbol{L} \cdot \boldsymbol{\Phi}(\boldsymbol{x})$ if the above $\boldsymbol{L}$ is column sparse. We also provide the following nuclear-norm based formulation to consider the more general case of low-dimensional space.

**Nuclear-Norm based Regularization.** To ensure the projection result $\boldsymbol{L} \cdot \boldsymbol{\Phi}(\boldsymbol{x})$ in a low-dimensional space, a more general way is directly restricting the projection matrix $\boldsymbol{L}$ to be low-rank. Then, the column vectors of $\boldsymbol{L}$ will be linearly dependent so that we can remove the redundant column to achieve a low-dimensional projection space. The realized formulation can be written as

$$\mathcal{R}_{\text{nuclear}}(\boldsymbol{\Phi}, \boldsymbol{L}) = \mathbb{E}_{\boldsymbol{x} \in \mathcal{X}}[\,\|\boldsymbol{L}^\top \boldsymbol{L} \cdot \boldsymbol{\Phi}(\boldsymbol{x}) - \boldsymbol{\Phi}(\boldsymbol{x})\|_2^2\,] + \alpha \|\boldsymbol{L}\|_*, \tag{7}$$

where $\boldsymbol{L} \in \mathbb{R}^{H \times H}$ and $\alpha > 0$ is tuned by users. When we obtain the learned projection matrix $\boldsymbol{L}^*$, we need to further compute its *maximal linearly independent set* $\mathcal{A}$, and then we calculate the final projection matrix $\widehat{\boldsymbol{L}} \in \mathbb{R}^{H \times H}$ by setting [1] the redundant columns of $\boldsymbol{L}^*$ to $\boldsymbol{0}$.

For the above two different formulations in Eq. (6) and Eq. (7), it is hard to say which one is theoretically better. Actually, their final performance may also be influenced by the non-convexity of the learning objectives. Therefore, in our experiments, we evaluate both the two regularizations on multiple domains. Now we want to summarize our final learning objective as follows.

**Learning Objective of CLLR.** Based on the realized formulation in Eq. (6) and Eq. (7), we can easily deploy the proposed two regularizers in the learning objective of conventional CL methods. Without loss of generality, for most existing CL methods equipped with NCE loss, we build the following framework of contrastive learning with low-dimensional reconstruction (CLLR)

$$\min_{\boldsymbol{\Phi} \in \mathcal{H}, \boldsymbol{L} \in \mathbb{R}^{H \times H}} \{\mathcal{F}(\boldsymbol{\Phi}, \boldsymbol{L}) = \mathcal{L}_{\text{NCE}}(\boldsymbol{\Phi}) + \lambda \mathcal{R}(\boldsymbol{\Phi}, \boldsymbol{L})\}, \tag{8}$$

where the regularization parameter $\lambda > 0$ is tuned by users and the regularizer $\mathcal{R}(\boldsymbol{\Phi}, \boldsymbol{L})$ can be realized by $\mathcal{R}_{2,1}(\boldsymbol{\Phi}, \boldsymbol{L})$ and $\mathcal{R}_{\text{nuclear}}(\boldsymbol{\Phi}, \boldsymbol{L})$ in Eq. (6) and Eq. (7), respectively. As a regularized learning objective, CLLR is very generic because here the loss term $\mathcal{L}_{\text{NCE}}(\boldsymbol{\Phi})$ can be implemented by many existing CL methods. In the next subsection, we provide iteration algorithm to solve Eq. (8).

### 2.3 Optimization

Minimizing the objective function in Eq. (8) is a typical batch optimization problem [48], where both the loss function $\mathcal{L}_{\text{NCE}}(\boldsymbol{\Phi})$ and the regularizer $\mathcal{R}(\boldsymbol{\Phi}, \boldsymbol{L})$ involve all training data. Therefore, we adopt the *stochastic gradient descent* (SGD) method [22] to solve it, and here we demonstrate the stochastic gradient for the objective function $\mathcal{F}(\boldsymbol{\Phi}, \boldsymbol{L})$. Specifically, for $n + 1$ (*i.e.*, the batch size) randomly selected data points $\{\boldsymbol{x}_{b_j} | \boldsymbol{x}_{b_j} \in \mathcal{X}, b_j \in B\}_{j=1}^{n+1}$, the NCE loss already has a stochastic form [2], so here we need to demonstrate the stochastic loss for the regularizer in the mini-batch, *i.e.*,

$$\mathcal{R}_B(\boldsymbol{\Phi}, \boldsymbol{L}) = [1/(n+1)] \sum_{i=1}^{n+1} \|\boldsymbol{L}^\top \boldsymbol{L} \cdot \boldsymbol{\Phi}(\boldsymbol{x}_{b_i}) - \boldsymbol{\Phi}(\boldsymbol{x}_{b_i})\|_2^2 + \alpha \widehat{\mathcal{R}}(\boldsymbol{L}), \tag{10}$$

---

[1] Here the $i$-column of $\widehat{\boldsymbol{L}}$ is $\widehat{\boldsymbol{L}}_i = \boldsymbol{L}_i^*$ if $\boldsymbol{L}_i^* \in \mathcal{A}$ and $\widehat{\boldsymbol{L}}_i = \boldsymbol{0}$ if $\boldsymbol{L}_i^* \notin \mathcal{A}$, in which $i = 1, 2, \ldots, H$.

[2] Here we denote NCE loss $\mathcal{L}_{\text{NCE}}(\boldsymbol{\varphi}) = \mathbb{E}[\ell(\boldsymbol{\varphi}; \{\boldsymbol{x}_{b_j}\}_{j=1}^{n+1})]$, where the function $\ell(\boldsymbol{\varphi}; \{\boldsymbol{x}_{b_j}\}_{j=1}^{n+1}) = -\log(\exp(\boldsymbol{\varphi}(\boldsymbol{x}_{b_{n+1}})^\top \boldsymbol{\varphi}(\boldsymbol{x}_{b_{n+1}}^+)) / (\exp(\boldsymbol{\varphi}(\boldsymbol{x}_{b_{n+1}})^\top \boldsymbol{\varphi}(\boldsymbol{x}_{b_{n+1}}^+)) + \sum_{j=1}^{n} \exp(\boldsymbol{\varphi}(\boldsymbol{x}_{b_j})^\top \boldsymbol{\varphi}(\boldsymbol{x}_{b_j}^-))))$. The index vector set $B = \{\boldsymbol{b} = (b_1, \ldots, b_{n+1})^\top | b_i, b_j = 1, \ldots, N, \; b_i \neq b_j, \; i, j = 1, \ldots, n+1\}$.

---

**Algorithm 1** Solving Eq. (8) via SGD.

---

**Input:** Training Data $\mathcal{X} = \{\boldsymbol{x}_i\}_{i=1}^N$; Step Size $\eta > 0$; Regularization Parameter $\lambda, \alpha > 0$; Batch Size $n \in \mathbb{N}_+$.
**Initialize:** Iteration Number $t = 0$.
**For** $t$ **from** 1 **to** $T$**:**

    1). Uniformly pick $(n+1)$ data points $\{\boldsymbol{x}_{b_j}\}_{j=1}^{n+1}$ from $\mathcal{X}$;

    2). Compute the gradient of $f(\boldsymbol{\Phi}; \{\boldsymbol{x}_{b_j}\}_{j=1}^{n+1}) = \ell(\boldsymbol{\Phi}; \{\boldsymbol{x}_{b_j}\}_{j=1}^{n+1}) + \lambda \mathcal{R}_B(\boldsymbol{\Phi}, \boldsymbol{L}; \{\boldsymbol{x}_{b_j}\}_{j=1}^{n+1})$ via Eq. (10):

    3). Update the learning parameters:

$$\boldsymbol{\Phi}_{(t+1)} \leftarrow \boldsymbol{\Phi}_{(t)} - \eta \nabla_{\boldsymbol{\Phi}} f(\boldsymbol{\Phi}, \boldsymbol{L}; \{\boldsymbol{x}_{b_j}\}_{j=1}^{n+1}) \quad \text{and} \quad \boldsymbol{L}_{(t+1)} \leftarrow \boldsymbol{L}_{(t)} - \eta \nabla_{\boldsymbol{L}} f(\boldsymbol{\Phi}, \boldsymbol{L}; \{\boldsymbol{x}_{b_j}\}_{j=1}^{n+1}), \quad (9)$$

**End.**
**Output:** The converged $\widetilde{\boldsymbol{\Phi}}$ and $\widetilde{\boldsymbol{L}}$.

---

where $\widehat{\mathcal{R}}(\boldsymbol{L})$ indicates the penalty $\|\boldsymbol{L}\|_{2,1}$ or $\|\boldsymbol{L}\|_*$ for different regularizations. Here we use the subgradients [2] of $\ell_{2,1}$-norm and nuclear-norm for optimization. Then the learning objective $\mathcal{F}(\boldsymbol{\Phi}, \boldsymbol{L})$ in Eq. (8) has the stochastic form $\ell(\boldsymbol{\Phi}; \{\boldsymbol{x}_{b_j}\}_{j=1}^{n+1}) + \lambda \mathcal{R}_B(\boldsymbol{\Phi}, \boldsymbol{L}; \{\boldsymbol{x}_{b_j}\}_{j=1}^{n+1})$. Based on such a stochastic loss, we further provide the SGD iteration steps in Algorithm 1 to solve Eq. (8).

In summary, introducing the projection layer (i.e., the projection matrix $\boldsymbol{L}$) merely incurs an additional stochastic gradient in Eq. (10). It means that our method can be easily implemented in most existing CL methods and only introduces very little computational overheads. In the next section, we prove that the iteration sequence $\boldsymbol{\Phi}_{(1)}, \ldots, \boldsymbol{\Phi}_{(T)}$ in Algorithm 1 converges to a stationary point of the learning objective $\mathcal{F}$ with a convergence rate $\mathcal{O}(1/\sqrt{T})$, where $T$ is the number of iterations.

## 3 Theoretical Analyses

In this section, we further provide in-depth theoretical analyses for our proposed method. We investigate the convergence of learning algorithm and the lower bound of min-max distance ratio to demonstrate the effectiveness of our method. All proofs are given in *supplementary materials*.

### 3.1 Convergence Analysis

As we described before, the learning objective of CLLR is a regularized empirical loss which is different from the traditional empirical loss solved by SGD, so here we provide careful convergence analysis for the SGD based iterations, *i.e.*, the Algorithm 1. Specifically, we suppose the learning objective has $\delta$-bounded gradient, and then we have the following Theorem 2.

**Theorem 2.** *If the function $\mathcal{F}(\boldsymbol{\Phi}, \boldsymbol{L})$ has $\delta$-bounded gradient (i.e., $\|\nabla \mathcal{F}(\boldsymbol{\Phi}, \boldsymbol{L})\|_2 < \delta$), then we let $\eta = \sqrt{2(\mathcal{F}(\boldsymbol{\Phi}_{(0)}, \boldsymbol{L}_{(0)}) - \mathcal{F}(\boldsymbol{\Phi}^*, \boldsymbol{L}^*))/(S\delta^2 T)}$, and for the iterations in Algorithm 1 we have that*

$$\min_{0 \le t \le T-1} \mathbb{E}[\|\nabla \mathcal{F}(\boldsymbol{\Phi}_{(t)}, \boldsymbol{L}_{(t)})\|_2] \le \sqrt{2S(\mathcal{F}(\boldsymbol{\Phi}_{(0)}, \boldsymbol{L}_{(0)}) - \mathcal{F}(\boldsymbol{\Phi}^*, \boldsymbol{L}^*))/T} \delta, \quad (11)$$

*where $S > 0$ is a lipschitz constant such that $\|\nabla \mathcal{F}(\boldsymbol{\Phi}, \boldsymbol{L}) - \nabla \mathcal{F}(\boldsymbol{\Phi}', \boldsymbol{L}')\|_2 \le S \|[\boldsymbol{\Phi}, \boldsymbol{L}] - [\boldsymbol{\Phi}', \boldsymbol{L}']\|_2.$*

The above Eq. (11) clearly reveals that the iteration results in Algorithm 1 can gradually converge to a stationary point with a convergence rate $\mathcal{O}(1/\sqrt{T})$ when setting the proper learning rate $\eta$ and increasing the iteration number $T$. Therefore, the convergence of our learning algorithm is guaranteed though the additional projection layer and regularization term are introduced.

### 3.2 Lower Bound of Min-Max Distance Ratio

Now, we further analyze the distance between pairwise instances in the low-dimensional space. As we mentioned before, in the high-dimensional space, the min-max distance ratio trends to be 0 and thus the distance function will lose its discriminatory. Therefore, we want to investigate the value of min-max distance ratio in low-dimensional feature space learned by our method.

Our method explicitly constrain the dimensionality of the feature space, so it is intuitive that the min-max distance ratio $(\mathcal{D}_{\boldsymbol{\Phi}}^{\max}(H) - \mathcal{D}_{\boldsymbol{\Phi}}^{\min}(H))/\mathcal{D}_{\boldsymbol{\Phi}}^{\min}(H)$ in Eq. (5) should certainly be lower-bounded. To be religious, we have the following Theorem 3 to reveal the lower bound of distance ratio.

**Theorem 3.** *For any given $n+1$ i.i.d. random data points $\boldsymbol{x}$, $\boldsymbol{x}_1$, $\boldsymbol{x}_2$, $\dots$, $\boldsymbol{x}_n \in \mathbb{R}^m$, we denote that $\mathcal{D}_{\widehat{\boldsymbol{\Phi}},\widehat{\boldsymbol{L}}}^{\max} = \max\{\mathcal{D}_{\widehat{\boldsymbol{\Phi}},\widehat{\boldsymbol{L}}}(\boldsymbol{x}, \boldsymbol{x}_i)|i=1,2,\dots,n\}$ and $\mathcal{D}_{\widehat{\boldsymbol{\Phi}},\widehat{\boldsymbol{L}}}^{\min} = \min\{\mathcal{D}_{\widehat{\boldsymbol{\Phi}},\widehat{\boldsymbol{L}}}(\boldsymbol{x}, \boldsymbol{x}_i)|i=1,2,\dots,n\}$, and then we have that*

$$\mathcal{P}\left\{(\mathcal{D}_{\widehat{\boldsymbol{\Phi}},\widehat{\boldsymbol{L}}}^{\max} - \mathcal{D}_{\widehat{\boldsymbol{\Phi}},\widehat{\boldsymbol{L}}}^{\min})/\mathcal{D}_{\widehat{\boldsymbol{\Phi}},\widehat{\boldsymbol{L}}}^{\min} \geq \alpha\lambda C(\mathcal{X})\right\} = 1, \tag{12}$$

*where $\mathcal{D}_{\widehat{\boldsymbol{\Phi}},\widehat{\boldsymbol{L}}}(\boldsymbol{x}, \boldsymbol{x}_i) = \|\widehat{\boldsymbol{L}}\widehat{\boldsymbol{\Phi}}(\boldsymbol{x}) - \widehat{\boldsymbol{L}}\widehat{\boldsymbol{\Phi}}(\boldsymbol{x}_i)\|_2/rank(\widehat{\boldsymbol{L}})$, and $\widehat{\boldsymbol{\Phi}}$ and $\widehat{\boldsymbol{L}}$ are learned from Eq. (8).*

From the above Eq. (12), we can easily observe that the min-max distance ratio has an explicit lower bound which is mainly determined by the two regularization parameters $\alpha$ and $\lambda$ (given the training data $\mathcal{X}$). It means that the low-rank reconstruction terms (*i.e.*, Eq. (6) and Eq. (7)) make the min-max distance ratio be controllable, and the larger regularization parameters can produce the better lower bound. When the min-max distance ratio is lower-bounded, our CLLR predicts low similarities for inter-cluster and high similarities for intra-cluster, so that the learned embedding effectively captures the intrinsic similarities / features and thus improving the performance of downstream tasks.

## 4 Experimental Results

In this section, we show experimental results on real-world datasets to validate the effectiveness of our proposed method. In detail, we first conduct ablation study to reveal the usefulness of our introduced new block and new regularizers. Then, we compare our proposed learning algorithm with existing state-of-the-art models on vision and language tasks. Finally, we test our method on the CL based reinforcement learning task. Further experiments such as parametric sensitivity and running time comparison are given in *supplementary materials*. The training process is implemented on Pytorch [29] with NVIDIA TeslaV100 GPUs. We adopt the projection result $\boldsymbol{L}\boldsymbol{\Phi}(\boldsymbol{x})$ for feature extraction, where regularization parameters $\lambda$ and $\alpha$ are fixed to $0.1$ and $10$, resepectively. The hyper-parameters of compared methods are set to the recommended values according to their original papers.

### 4.1 Ablation Study

In this subsection, we conduct ablation study on the superiority of the low-dimensional contrastive embedding (*i.e.*, our method) over the traditional contrastive embedding (*i.e.*, the baseline method). We use the *STL-10* and *CIFAR-10* datasets to train the baseline *SimCLR* [7] and two implementations of CLLR, *i.e.*, the $\ell_{2,1}$-norm based regularization and nuclear-norm based regularization. We train all models with 100 and 400 epochs with the

Table 1: Classification accuracy rates (mean $\pm$ std) of high-dimensional embedding and low-dimensional embedding on *STL-10* and *CIFAR-10* datasets (negative sample size = 256).

| METHOD | STL-10 | | CIFAR-10 | |
|---|---|---|---|---|
| | epochs=100 | epochs=400 | epochs=100 | epochs=400 |
| 4096-dim. (w/o $\mathcal{R}(\boldsymbol{\Phi}, \boldsymbol{L})$) | $55.1 \pm 1.1$ | $75.2 \pm 3.1$ | $65.1 \pm 1.9$ | $85.4 \pm 4.2$ |
| 3072-dim. (w/o $\mathcal{R}(\boldsymbol{\Phi}, \boldsymbol{L})$) | $54.4 \pm 3.1$ | $75.2 \pm 2.1$ | $67.2 \pm 3.5$ | $86.9 \pm 6.1$ |
| 2048-dim. (w/o $\mathcal{R}(\boldsymbol{\Phi}, \boldsymbol{L})$) | $56.3 \pm 2.1$ | $76.2 \pm 1.1$ | $66.3 \pm 3.1$ | $89.3 \pm 2.1$ |
| 512-dim. (w/o $\mathcal{R}(\boldsymbol{\Phi}, \boldsymbol{L})$) | $56.4 \pm 2.5$ | $75.2 \pm 0.1$ | $66.4 \pm 5.1$ | $90.3 \pm 0.6$ |
| 256-dim. (w/o $\mathcal{R}(\boldsymbol{\Phi}, \boldsymbol{L})$) | $55.3 \pm 4.1$ | $74.2 \pm 2.1$ | $64.3 \pm 5.1$ | $88.3 \pm 3.1$ |
| 512-dim. (w/o sparity, $\alpha = 0$) | $\boldsymbol{56.5 \pm 2.5}$ | $75.5 \pm 0.5$ | $66.2 \pm 4.9$ | $90.1 \pm 1.2$ |
| 256-dim. (w/o sparity, $\alpha = 0$) | $55.9 \pm 2.1$ | $74.1 \pm 2.3$ | $64.7 \pm 2.1$ | $88.4 \pm 2.6$ |
| 512-dim. (w / $\ell_{2,1}$-norm) | $56.3 \pm 8.2 -$ | $78.3 \pm 0.5 \checkmark$ | $\boldsymbol{67.5 \pm 0.2} -$ | $\boldsymbol{92.5 \pm 0.2} \checkmark$ |
| 512-dim. (w / nuclear-norm) | $56.2 \pm 3.2 -$ | $\boldsymbol{79.2 \pm 0.2} \checkmark$ | $67.5 \pm 2.5 -$ | $92.5 \pm 2.3 \checkmark$ |
| 256-dim. (w / $\ell_{2,1}$-norm) | $56.2 \pm 1.2 -$ | $\boldsymbol{79.3 \pm 0.5} \checkmark$ | $65.5 \pm 0.5 -$ | $92.3 \pm 0.3 \checkmark$ |
| 256-dim. (w / nuclear-norm) | $56.3 \pm 3.2 -$ | $79.2 \pm 0.2 \checkmark$ | $65.2 \pm 5.5 -$ | $\boldsymbol{93.1 \pm 1.3} \checkmark$ |

same batch size and learning rate, respectively, and we record the test accuracy of all methods by fine-tuning a linear *softmax*. The baseline method learns contrastive embeddings in the high-dimensional space (dimension = 2048, 3072, and 4096) and the simply fixed low-dimensional space (dimension = 256 and 512). We also include the baseline results that do not use the $\ell_{2,1}$-norm and nuclear norm constraints (*i.e.*, $\alpha = 0$). Our method learns embeddings in low-dimensional space, where we use the regularizer to maintain the corresponding non-zero columns in the projection matrix $\boldsymbol{L}$.

We record the test accuracy (mean $\pm$ std, 5 random trials) of compared methods at the 100-*th* epoch and 400-*th* epoch in Tab. 1. We can observe that the baseline method is better than our method in the first 100 epochs, but the two implementations of our method can outperform the baseline method with the increase of iterations. This is because that the baseline method only emphasizes on the instance discrimination, so it can quickly discriminate the training data in the early epochs. However, in the latter epochs, the low-rank reconstruction in our method becomes useful in capturing the similarity between pairwise instances. Meanwhile, we can find that the average accuracy of nuclear-norm based

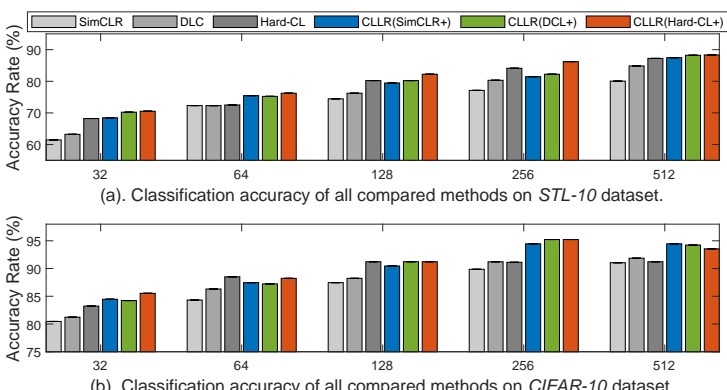

(a). Classification accuracy of all compared methods on *STL-10* dataset.

(b). Classification accuracy of all compared methods on *CIFAR-10* dataset.

Figure 4: Classification accuracy of all methods on *STL-10* and *CIFAR-10* datasets. The negative sample size is from 32 to 512.

regularization is slightly higher than the $\ell_{2,1}$-norm based one on both two datasets. Furthermore, we also perform the $t$-test at significance level $0.05$ in the last column, and "✓" indicates that our method is significantly better than the best baseline result. In our following experiments, we employ the 256-dimensional latent features for multiple domain tasks.

## 4.2 Experiments on Sentence Representation

In this subsection, we employ the *BookCorpus* dataset [23] to evaluate the performance of all compared methods on six text classification tasks, including movie review sentiment (*MR*), product reviews (*CR*), subjectivity classification (*SUBJ*), opinion polarity (*MPQA*), question type classification (*TREC*), and paraphrase identification (*MSRP*). We follow the experimental settings in the baseline method *quick-thought* (QT) [26], which chooses the neighboring sentences as positive pairs. Here the 10-fold cross validation is adopted, and the average classification accuracy is listed in Tab. 2.

For the six classification tasks, our method improves the classification accuracy of baseline method QT for at least one percentage on most classification benchmarks. The distance histograms of QT, *debiased contrastive learning* (DCL) [11], *hard negative based contrastive learning* (HCL) [30], and our CLRR are

Table 2: Classification accuracy (%) of all methods on *Book-Corpus* dataset including six text classification tasks.

| METHOD | MR | CR | SUBJ | MPQA | TREC | MSRP |
|---|---|---|---|---|---|---|
| QT[26] | 76.8 | 81.3 | 86.6 | 93.4 | 89.8 | 73.6 |
| DCL[11] | 76.2 | 82.9 | 86.9 | 93.7 | 89.1 | 74.7 |
| HCL[30] | 77.4 | 83.6 | 86.8 | 93.4 | 88.7 | 73.5 |
| CLLR(DCL+$\ell_{2,1}$-norm) | 77.9 | 83.3 | **87.9** | 93.7 | **91.3** | 75.2 |
| CLLR(DCL+nuclear-norm) | **78.2** | **83.7** | 87.2 | **95.8** | 91.2 | **75.7** |

shown in Fig. 3. We clearly observe that our method obtains the more accurate distance determination than baseline methods, and this reveals that our method is effective for the text classification task.

## 4.3 Experiments on Image Classification

In this subsection, we validate the effectiveness of our method on the image classification task. Here we select *contrastive multiview coding* (CMC) [35] as baseline methods, and implement our method CLLR under such a classical framework. We also compare our method with three additional state-of-the-art methods including DCL, HCL, SwAV [3], and CO2 [38] on *STL-10* [12], *CIFAR-10* [24], and *ImageNet-100* [31] datasets. All methods are fairly implemented by the *ResNet50* with the same training epoch 100.

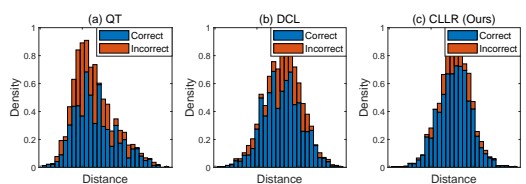

Figure 3: Distance histograms obtained by different methods (QT, DCL, and our proposed CLRR) on *Book-Corpus* dataset. The proportion of incorrect prediction of CLLR is clearly lower than the compared methods.

For *STL-10* and *CIFAR-10* datasets, we record the classification accuracy of all compared methods with varying numbers of negative sample.

From Fig. 4, we can clearly observe that our method CLLR successfully improves the baseline for at least $1\%$ and $2\%$ on *CIFAR-10*

dataset and *STL-10* dataset, respectively. Similar experiments are conducted on *ImageNet-100* dataset, and Tab. 3 shows that our method consistently improves all baseline methods, where our method improves the baseline CMC from 73.58% to 76.91%. For different negative sample sizes, the accuracy rates of our method are also higher than all compared methods, and it clearly demonstrates the effectiveness of our method. Since CLLR is implemented on different baselines, our method has good compatibility with existing CL algorithms on the image classification task. In *supplementary materials*, we further compare our method with the distillation based CL models [5, 47] (*i.e.*, the low-dimensional small networks supervised by the original contrastive embeddings), and the results clearly demonstrate the superiority of our method.

## 4.4 Experiments on Reinforcement Learning

This subsection further extends our experiments on reinforcement learning task, which is another application scenario of contrastive learning. Here the *contrastive unsupervised representations for reinforcement learning* (CURL) [25] method is employed to perform image-based policy control on representation learned by the CL algorithm. All methods are tested on the DeepMind control suite [34], which consists of six control tasks listed in Tab. 4. By following the experimental settings in CURL, the positive pair is built by simply cropping a single image, and the negative pair is composed of each two images in the control sequence. All methods are retrained for 3 times, and the corresponding means and standards of 100K scores are shown in Tab. 4.

Table 3: Classification accuracy (%) of all methods on *ImageNet-100* dataset with negative sample size 1024 and 4096.

| METHOD | 1024 | | 4096 | |
|---|---|---|---|---|
| | Top1 | Top5 | Top1 | Top5 |
| CMC[35] | 60.23 | 79.23 | 73.58 | 92.06 |
| SwAV[3] | 60.93 | 79.43 | 75.78 | 92.86 |
| DCL[11] | 61.01 | 78.99 | 74.60 | 92.08 |
| HCL[30] | 60.89 | 79.33 | 74.66 | 92.32 |
| CO2[38] | 61.21 | 79.32 | 73.96 | 93.02 |
| CLLR(CMC+$\ell_{2,1}$-norm) | 62.03 | 80.64 | 75.97 | 94.22 |
| CLLR(CMC+nuclear-norm) | 61.23 | 80.50 | **76.91** | 94.03 |
| CLLR(HCL+$\ell_{2,1}$-norm) | 61.29 | **81.10** | 76.88 | 94.19 |
| CLLR(HCL+nuclear-norm) | **62.43** | 80.98 | 76.89 | **94.25** |

For the six control tasks, our method consistently outperforms the baseline method CURL with higher means. When compared to DCL and HCL methods, our method almost achieves the best results in all six scenarios. Although our method CLRR (CURL+nuclear-norm) has slightly lower scores than CURL or DCL on the *Run / Walk* tasks, our method shows smaller variance. Moreover, when

Table 4: 100K Scores (mean $\pm$ std, 3 random trials) achieved by all methods on the six control tasks.

| METHOD | *Spin* | *Swingup* | *Easy* | *Run* | *Walk* | *Catch* |
|---|---|---|---|---|---|---|
| CURL[25] | 413±53 | 680±32 | 908±86 | **298±38** | 621±121 | 826±42 |
| DCL[11] | 422±23 | 672±52 | 878±96 | 248±98 | **626±98** | 836±12 |
| HCL[30] | 420±61 | 678±82 | 869±116 | 268±42 | 623±26 | 819±62 |
| CLLR(CURL+) | **424±53** | 683±23 | **925±33** | 296±32 | 625±23 | 843±17 |
| CLLR(DCL+) | 423±13 | **684±83** | 919±57 | 287±67 | 625±33 | **844±27** |
| CLLR(HCL+) | 422±41 | 681±13 | 911±85 | 292±78 | **626±59** | 839±33 |

we incorporate our method to DCL and HCL, our method could further improve the overall scores of compared methods on the six tasks. This also reveals that our method is compatible with existing CL algorithms on the reinforcement learning task.

## 5 Conclusion and Future Work

In this paper, we considered the issue of high-dimensional features existing in the current contrastive learning method. To overcome such an issue, we proposed a novel framework called contrastive learning with low-dimensional reconstruction (CLLR), which uses a sparse projection layer to reduce the dimensionality of the feature embedding. We reconstructed the original high-dimensional features in the low-dimensional projection space while preserving the basic objective for instance discrimination, and thus successfully learning low-dimensional contrastive embeddings. To the best of our knowledge, this is the first work in CL that considers reducing the feature dimensionality. We conducted intensive theoretical analyses to guarantee the effectiveness of our method. Comparison experiments on real-world datasets across multiple domains indicated that our learning algorithm acquires more reliable feature embedding than state-of-the-art methods. Both the theoretical and experimental results clearly demonstrated the necessity / significance of learning low-dimensional contrastive embeddings. Our approach mainly focuses on the mainstream CL models which use both positive and negative pairs. The effectiveness of negative-free CL has also been shown by recent works such as BYOL and SimSiam. When the negative pairs are unavailable, exploring the corresponding optimal (low-dimensional) projection space would be interesting future work.

**Acknowledgment**

S.C., G.N., and M.S. were supported by JST AIP Acceleration Research Grant Number JPMJCR20U3, Japan. M.S. was also supported by the Institute for AI and Beyond, UTokyo.

C.G., J.L., and J.Y. were supported by NSF of China (Nos: U1713208, 61973162, 62072242), NSF of Jiangsu Province (No: BZ2021013), NSF for Distinguished Young Scholar of Jiangsu Province (No: BK20220080), and the Fundamental Research Funds for the Central Universities (Nos: 30920032202, 30921013114).

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
