# OpenReview forum: "Learning Contrastive Embedding in Low-Dimensional Space"
_NeurIPS.cc/2022/Conference — NeurIPS 2022 Accept_

### Official Review · Reviewer_VTWD · 2022-06-30

**Rating:** 7
**Confidence:** 3
**Soundness:** 3 good
**Presentation:** 3 good
**Contribution:** 3 good

**Summary:**

This work proposes a framework (CLLR) to learn embeddings through contrastive learning (CL). The proposed approach learns a low dimensional projection of the feature space used for CL. Furthermore, theoretical bound is provided for CLLR along with empirical explorations using the learned low-dimensional embedding.

**Questions:**

As mentioned earlier, a central question remains regarding the whole approach. The low dimensional embeddings can directly be learned through CL and the proposed regularizations. Therefore, learning embeddings through CL followed by a projection layer seems like added complexity. Perhaps an explanation on why the low dimensional embeddings cannot be learned directly might be useful.

In the empirical explorations, were the models trained using both CL and the downstream task based loss (like multi-task learning) ?

**Limitations:**

The authors do touch upon limitations of their approach where others approaches can potentially work better.

**Strengths And Weaknesses:**

Originality :

The work is somewhat original. The authors combine well known techniques in manner that produces a useful framework for producing low dimensional features through CL

Quality :

This work proposes a framework that is decently analyzed through theoretical bounds and empirical explorations using multiple diverse tasks. The authors also provide ablation studies to strengthen the proposed framework. However, a central question still remains. Why is it that the low dimensional embedding cannot be learned directly through CL. The proposed regularizations can also be applied to the final matrix that produces the features. The need for learning the CL based embeddings followed by a projection layer seems to add complexity that perhaps could have been avoided.


Clarity:

This work follows a structured approach and explains the procedure in a clear manner.

Significance :

The concept of low dimensional embeddings learned through CL is very useful. This work provides a useful step in that direction.

---

> ### Author Response · Authors · 2022-08-02
> **Response to Reviewer VTWD**
>
> Thank you for your positive and constructive comments! Our explanation and clarification can be found as follows.
>
> ---
>
> **Comment 1:** Why is it that the low dimensional embedding cannot be learned directly through CL?
>
> **Response 1:** As we discussed in Section 1, simply reducing the dimensionality of the output layer can indeed obtain a low-dimensional feature embedding. However, such a directly reduced low-dimensional embedding cannot achieve the goal of instance discrimination (encouraging using high-dimensional feature space to discriminate instances). Existing works [Jing et al., ICLR’22] also reveal that directly using low-dimensional space will cause the dimensional collapse with insufficient instance discrimination. Therefore, we propose a new CL framework to solve this dilemma, where we maintain the instance discrimination in high-dimensional space while building the new sparse projection layer to reconstruct the high-dimensional features with a new sparse-regularized reconstruction loss. Our experiments also demonstrate that introducing such a sparse projection layer and the corresponding reconstruction loss can significantly improve the recognition performance of several baseline methods on multiple domain tasks. We have also added the corresponding explanations in Section 2.2 of the revised paper to further clarify this point.
>
> ---
>
> **Comment 2:** The projection layer seems to add complexity.
>
> **Response 2:** Yes, we agree with the reviewer that the new projection layer may increase the computational complexity. However, we would like to clarify that such additional complexity is independent to the size of training data, so that the practical training complexity is almost the same as the baseline method. Here we provide (as we had also shown in the appendix) the following table to show the training time comparison (using 2 TeslaV100 GPUs, with 100 epochs and batch_size = 512/1024, in hours), where we observe that the proposed regularizer only adds very little additional time consumption. Therefore, we believe the additional complexity is negligible for practical use.
> | Method &nbsp; | &nbsp; CIFAR-10_#512 &nbsp; | &nbsp; CIFAR-10_#1024 &nbsp; | &nbsp; ImageNet-100_#512 &nbsp; | &nbsp; ImageNet-100_#1024 &nbsp; |
> | :--- | :----: | :----: | :----: | :----: |
> | SimCLR | 2.3 | 1.3 | 10.9 | 5.5 |
> | DCL | 2.5 | 1.4 | 11.2 | 5.7 |
> | SimCLR+CLLR | 2.4 | 1.5 | 11.2 | 5.8 |
> | DCL+CLLR | 2.6 | 1.6 | 11.5 | 5.9 |
>
> ------
>
> **Comment_3:** Were the models trained using both CL and the downstream task based loss (like multi-task learning)?
>
> **Response_3:** As the reviewer considered, the baseline methods and our proposed method did in fact use additional loss functions for downstream tasks. However, this is a little different from multi-task learning. As we know, multi-task learning seeks to learn to perform multiple recognition tasks simultaneously, yet CL aims to learn a generic feature representation for subsequent downstream tasks. Accordingly, multi-task learning usually involves concurrently learning models with several integrated objectives and simultaneously minimizing the integrated loss functions. In CL, we learn the feature representation by only minimizing the single self-supervision based learning objective, and thus the loss function of CL and the loss functions of the corresponding downstream tasks are minimized in different phases (*i.e.*, pre-training and fine-tuning).

---

> > ### Comment · Reviewer_VTWD · 2022-08-07
> > **Reply to the Author's Response**
> >
> > Thanks for the clarifications. In the end the feature size has a significant influence. Perhaps there is an appropriate size between the high dimensional baseline values (such as 2048 and 4096)  and the low sizes (such as 256 and 512) that could be sufficient for use with common CL setups.

---

> > > ### Author Response · Authors · 2022-08-08
> > > **Thank you for the reply**
> > >
> > > Dear Reviewer VTWD,
> > >
> > > Thank you very much for your timely reply!
> > >
> > > > In the end the feature size has a significant influence. Perhaps there is an appropriate size between the high dimensional baseline values (such as 2048 and 4096) the low sizes (such as 256 and 512) that could be sufficient for use with common CL setups.
> > >
> > > Yes, we agree with the reviewer that the feature dimension (size) usually affects both training and test errors, and this phenomenon is also observed in some existing CL methods such as SimCLR [Chen et al., ICML'20] and SimSiam [Chen et al., CVPR'21].
> > >
> > > We also agree with the reviewer that there should be an intermediate dimension value that could achieve acceptable performance for common CL methods. However, such an intermediate dimension is actually a *tradeoff* between the training objective (*i.e.*, the instance discrimination) and the generalization ability (*i.e.*, avoiding the curse of dimensionality to obtain reliable similarity predictions), so there is still room for improving the final recognition accuracy, which is co-determined by both the training effectiveness and generalization ability. In comparison, our method employs different layer parameters for model training and dimensionality reduction, respectively, so that we can reach the target of instance discrimination and avoid the curse of dimensionality *simultaneously*. Although we still have a hyperparameter $\alpha$ to control the layer sparsity, it can be easily tuned and is rather stable to the final performance (as we shown in Tab 0.1/0.2 in the appendix).
> > >
> > > To be more rigorous, here we further provide experiments to list both the training and test errors of the baseline CL method when we change the feature dimension from 256 to 2048 (with smaller intervals). In the following table, we record the training/test errors of the baseline method SimCLR (with different feature dimensions) on *STL-10* and *CIFAR-10* datasets, where we use a similar setting as in Section 4.1.
> > >
> > > | Method | &nbsp; &nbsp; STL-10_#400-eps &nbsp; | &nbsp; &nbsp; CIFAR-10_#400-eps &nbsp; |
> > > | :--- | :---- | :---- |
> > > | 2048-dim. w/o $\mathcal{R} (\boldsymbol{\varPhi}, \boldsymbol{L})$ |  &nbsp; &nbsp;  2.6±2.2 / 23.8±1.1 | &nbsp; &nbsp;  2.5±1.6 / 10.7±2.1 |
> > > | 1792-dim. w/o $\mathcal{R} (\boldsymbol{\varPhi}, \boldsymbol{L})$ |  &nbsp; &nbsp;  2.6±0.2 / 25.2±2.1 | &nbsp; &nbsp;  2.6±2.6 / 10.7±2.1 |
> > > | 1536-dim. w/o $\mathcal{R} (\boldsymbol{\varPhi}, \boldsymbol{L})$ |  &nbsp; &nbsp;  3.5±0.8 / 24.9±1.1 | &nbsp; &nbsp;  3.1±1.1 / 11.7±1.1 |
> > > | 1280-dim. w/o $\mathcal{R} (\boldsymbol{\varPhi}, \boldsymbol{L})$ |  &nbsp; &nbsp;  4.3±1.1 / 25.9±2.1 | &nbsp; &nbsp;  3.5±0.6 / 11.7±2.1 |
> > > | 1024-dim. w/o $\mathcal{R} (\boldsymbol{\varPhi}, \boldsymbol{L})$ |  &nbsp; &nbsp;  5.5±2.1 / 26.1±2.1 | &nbsp; &nbsp;  4.1±1.1 / 11.2±0.9 |
> > > | 896-dim. w/o $\mathcal{R} (\boldsymbol{\varPhi}, \boldsymbol{L})$ |  &nbsp; &nbsp;  6.8±2.2 / 25.9±1.8 | &nbsp; &nbsp;  4.5±1.2 / 10.9±3.2 |
> > > | 768-dim. w/o $\mathcal{R} (\boldsymbol{\varPhi}, \boldsymbol{L})$ |  &nbsp; &nbsp;  7.3±2.2 / 25.9±1.9 | &nbsp; &nbsp;  4.9±1.2 / 10.8±3.1 |
> > > | 640-dim. w/o $\mathcal{R} (\boldsymbol{\varPhi}, \boldsymbol{L})$ |  &nbsp; &nbsp;  9.1±1.5 / 25.8±1.1 | &nbsp; &nbsp;  5.1±1.6 / 10.7±2.1 |
> > > | 512-dim. w/o $\mathcal{R} (\boldsymbol{\varPhi}, \boldsymbol{L})$ |  &nbsp; &nbsp;  9.2±1.2 / 24.8±0.1 | &nbsp; &nbsp;  5.7±1.6 / 9.7±0.6 |
> > > | 256-dim. w/o $\mathcal{R} (\boldsymbol{\varPhi}, \boldsymbol{L})$  &nbsp;  &nbsp;  &nbsp; | &nbsp; &nbsp;  10.2±2.0 / 25.8±2.1 | &nbsp; &nbsp;  8.2±2.1 / 11.7±3.1 |
> > > | 512-dim. w/ nuclear (ours)  &nbsp; &nbsp; | &nbsp; &nbsp;   **3.8±1.2 / 20.8±0.2** ✔ | &nbsp; &nbsp;  **2.5±1.2 / 7.5±2.3** ✔ |
> > > | 256-dim. w/ nuclear (ours)  &nbsp; &nbsp; | &nbsp; &nbsp;   **4.2±0.1 / 20.8±0.2** ✔ | &nbsp; &nbsp;  **2.1±1.1 / 6.9±1.3** ✔ |
> > >
> > > We can observe that the 512-dimensional feature obtains the best results on the two datasets, so we think the 512-dimension is an appropriate choice to make a tradeoff for the baseline CL method. In this case, our method can still significantly improve the best results of the baseline on the two datasets. This is because our method simultaneously achieves the training objective and meanwhile avoids the curse of dimensionality. Therefore, it is really useful and critically important to introduce our new sparse projection layer and the corresponding new regularizer for learning low-dimensional contrastive embeddings.
> > >
> > > Thanks a lot again!
> > >
> > > Best regards,
> > >
> > > Authors of Paper ID 5171

---

### Official Review · Reviewer_4xAW · 2022-07-02

**Rating:** 4
**Confidence:** 5
**Soundness:** 3 good
**Presentation:** 2 fair
**Contribution:** 2 fair

**Summary:**

This paper proposes a novel regularization method for contrastive representation learning. The main idea is to introduce an additional layer that projects the representation to a lower-dimensional space. The additional projection layer is trained by minimizing the reconstruction loss where sparse regularization is applied. The projected (relatively) lower-dimensional representation is shown to be more effective in multiple downstream tasks.

**Questions:**

* Is the representation space constrained to a hypersphere or not?
* Does it assume a particular type of objective functions to be regarded as "contrastive learning"?
* Is there any reason that the proposed method is not applicable to non-contrastive classifiers?
* It would be informative if we can separate the contributions from reconstruction loss and sparsity regularization. What if we use reconstruction loss but without sparse regularization? There is no row in Table 1 corresponding to this setting.

**Limitations:**

I don't see any critical limitation of this work regarding negative social impact.

**Strengths And Weaknesses:**

I vote for rejection mainly due to the lack of clarity and the lack of good explanation. Meanwhile, I do find the experimental results compelling.

Strengths:

* Contrastive representation learning is an important field of machine learning and is a relevant topic.
* The proposed algorithm seems novel.
* The experiments are fairly extensive and the results are impressive. The proposed method is tested across multiple tasks, including image classification, text classification, and reinforcement learning, showing consistent improvements. Also, some of the results are provided with confidence intervals from repeated runs.
* I appreciate the significance of the empirical finding in this work.


Weaknesses:
* There is a large room for improvement in clarity. The paper needs to clearly define what contrastive learning is and what is not. Even though there is no single definition that is accepted across the whole community, at least the authors need to clarify what class of algorithm they are focusing on. This unclarity raises several questions during reading. Is the representation space constrained to a hypersphere or not? Does it assume a particular type of objective functions to be regarded as "contrastive learning"? Is there any reason that the proposed method is not applicable to non-contrastive classifiers? I will post this questions again on "Questions" section of the review.
* I am not persuaded by the argument made in Section 2.1 Motivation in general. As far as I know, many contrastive learning algorithms assume a hyperspherical latent space, so there is no point of comparing the volumes of a high-dimensional cube and a sphere.
* The argument from Theorem 1 is only indirectly relevant at best. In classification, what matters is the ordering of distances, not the exact values of distances. There is not direct evidence that high dimensionality hurts the quality of representation.

---

> ### Author Response · Authors · 2022-08-02
> **Response to Reviewer 4xAW (part 2/2)**
>
> **Comment_5:** Separating reconstruction loss and sparsity constraint.
>
> **Response_5:** Here we follow the reviewer’s suggestion to investigate the performance of our method discarding the sparsity constraint on *STL-10* and *CIFAR-10* datasets. In the following table, we can observe that merely using the single reconstruction loss without sparsity constraint hardly improves baselines in the first two rows (400 epochs, batch_size = 256). This is because the single reconstruction loss may incur the trivial solution, where the projection matrix $\boldsymbol{L}$ is learned to be the identity matrix, and this will not change the original feature, so that the final performance is almost the same. This implies that both the sparsity constraint and the reconstruction loss are critically important to the proposed method.
> | Method &nbsp; &nbsp; &nbsp; | &nbsp; STL-10 &nbsp; | &nbsp; CIFAR-10 &nbsp; |
> | :--- | :----: | :----: |
> | 512-dim. w/o $\mathcal{R} (\boldsymbol{\varPhi}, \boldsymbol{L})$ | 75.2±0.1 | 90.3±0.6 |
> | 256-dim. w/o $\mathcal{R} (\boldsymbol{\varPhi}, \boldsymbol{L})$  | 74.2±2.1 | 88.3±3.1 |
> | 512-dim. w/o sparsity constraint ($\alpha=0$) &nbsp; | 75.5±0.5 | 90.1±1.2 |
> | 256-dim. w/o sparsity constraint ($\alpha=0$) &nbsp; | 74.1±2.3 | 88.4±2.6 |
> | 512-dim. w/ sparsity (nuclear) | **79.2±0.2** | 92.5±2.3 |
> | 256-dim. w/ sparsity (nuclear) | 79.2±0.3 | **93.1±1.3** |
>
> We have added the above results in the revised paper.
>
> ---
>
> Thanks a lot again for your insightful and constructive comments! Hope our clarification could address your concerns. We would be grateful if the reviewer could provide more suggestions to further improve the clarity of our paper.

---

> ### Author Response · Authors · 2022-08-02
> **Response to Reviewer 4xAW (part 1/2)**
>
> Thank you for your insightful comments! We have tried our best to address all your concerns.
>
> ---
>
> **Comment_1:** Clarifying the specific CL algorithm that this paper focused on.
>
> **Response_1:** Thanks for the suggestion! We agree with the reviewer that there is still no single definition for CL that is accepted across the whole community. However, in our abstract/introduction, we had already declared that our paper focuses on the *instance discrimination* based CL methods, which aim to enlarge pairwise distances between every two instances. Such a principle of instance discrimination has also become the basic learning objective of most existing CL methods.
>
> To be more specific, as we discussed in Section 1.1/2.2, our method is implemented on the widely used NCE loss (the formulation is given in the footnote of the 3rd page), which considers not only the instance discrimination (building negative pairs) but also the data augmentation (generating positive pairs). Here we follow the reviewer’s suggestion, moving the formulation of NCE loss to the main body to clarify this point.
>
> ---
>
> **Comment_2:** The analysis in Section 2.1 for hyperspherical space. Is the representation space constrained to the hyperspherical space?
>
> **Response_2:** We want to clarify that the analysis in Section 2.1 still holds for the hyperspherical space which normalizes the final feature vector.
>
> First, as the feature embedding is assumed in the high-dimensional space, the unnormalized feature vectors tend to distribute in hypercube corners (due to the small volume ratio in Eq. (3)), making the norms of those feature vectors very close (*e.g.*, norms of corner vectors in the $H$-dimensional hypercube are all $\sqrt{H}/2$). Normalizing such norm-close feature vectors will almost not change the pairwise distances, so data points will still be far away from each other, leading to unreliable similarities.
>
> Second, to be more rigorous, as Theorem 1 does not have additional assumptions on the distance function and the corresponding feature embedding (no matter whether the feature vector is normalized or not), such a theorem still holds for those CL methods that assume a hyperspherical space. The small distance ratio in Eq. (5) reveals that the feature embedding hardly captures the underlying similarity between pairwise instances.
>
> In summary, our motivation in Section 2.1 is solid for instance discrimination based CL methods no matter whether they assume a hyperspherical space or not.
>
> Finally, for image/text related tasks (*i.e.*, baselines and our method in Section 4.1/4.2/4.3), we constrain the representation in the hyperspherical space. For the reinforcement learning task (*i.e.*, baselines and our method in Section 4.4), the representation is not constrained, as we find that the feature norms can preserve important scale information which improves the recognition performance.
>
> ---
>
> **Comment_3:** Theorem 1 does not consider the ordering (relative) distance.
>
> **Response_3:** It seems that there are some factual misunderstandings here. We would like to clarify that the conclusion of Theorem 1 *actually considers* the ordering distance, where Eq. (5) investigates the *relative ratio* between the minimal and maximal distances (but not the absolute distance value itself). Such a small relative ratio reveals that the minimal distance and maximal distance tend to be the same, so that the learned high-dimensional representation can hardly distinguish the inter-cluster and intra-cluster from each other (*i.e.*, harming the representation quality).
>
> ---
>
> **Comment_4:** Why is the method not applicable to non-contrastive approaches?
>
> **Response_4:** As we replied to the reviewer’s Comment_1, our method focuses on the instance discrimination based CL algorithms that consider negative pairs, because the curse of dimensionality is caused by discriminating such a large number of negative pairs. In our experiments, we implement our method on CL models that use both positive and negative pairs, but our proposed CLLR can also work with non-contrastive approaches (*i.e.*, negative-free methods, as we also replied to *Reviewer eAPH*).
>
> Here we conduct experiments on two representative negative-free CL baselines (BYOL [Grill et al., NeurIPS’20] and SimSiam [Chen et al., CVPR’21], merely using positive pairs) to validate the effectiveness of our proposed method. As shown in the following table, our method can consistently improve the compared methods upon themselves (TOP1 and TOP5 accuracy on *ImageNet-100* with 500 training epochs and batch_size = 1024/4096). We have added the additional results to the revised appendix.
> | Method | #1024_TOP1 | #1024_TOP5 | #4096_TOP1 | #4096_TOP5 |
> | :--- | :----: | :----: | :----: | :----: |
> | BYOL | 61.3 | 91.8 | 74.9 | 91.9 |
> | SimSiam | 70.9 | 91.9 | 73.6 | 92.8 |
> | CLLR+BYOL | 63.1 | 92.7 | **76.5** | 93.0 |
> | CLLR+SimSiam | **72.2** | **92.9** | 75.8 | **93.8** |

---

### Official Review · Reviewer_eAPH · 2022-07-11

**Rating:** 8
**Confidence:** 5
**Soundness:** 4 excellent
**Presentation:** 3 good
**Contribution:** 4 excellent

**Summary:**

This paper focuses on the issue of high dimensionality in the existing contrastive learning (CL) approaches. As CL usually encourages learning high-dimensional features to discriminate data points as much as possible, it may make the data points sparsely distributed in the high-dimensional feature space. In this case, the authors argue that the learning algorithm can hardly capture the underlying similarity between pairwise instances. Then the authors propose a new sparse regularized projection layer to address this problem, reconstructing the learned features in the low-dimensional (sparse) projection space and meanwhile maintaining the conventional instance discrimination of CL in the original high-dimensional space. The generic sparse/low-rank regularized loss functions are also designed with providing the corresponding optimization algorithm. Some theoretical analyses (Theorem 1, 2, and 3) are explored to demonstrate the motivation and effectiveness of the proposed method. Finally, experiments on multiple domains including image, text, and reinforcement datasets validate the superiority of the method.

**Questions:**

I have some more questions regarding the details of this work.

1).  What is the dimensionality of the finally used embedding for fine-tuning (with softmax)? It seems that the reconstructed features are still high-dimensional. If they use the low-dimensional latent features, this should be clarified if this paper is accepted.

2). I find that the authors omit some details in Eq. (3). It is unclear to us that why $\lim_{H\rightarrow\infty}(\pi^{H/2}/(H\cdot\Gamma(H/2)))/2^{H-1}\leq\lim_{H\rightarrow\infty}\pi^{(H-1)/2}/2^{H-1}$ can hold. I think this is critically import to the motivation, so please try to explain this more carefully.

3). In the optimization section, the authors use the classical SGD for model training. Is that possible if we try to use some more popular optimizers such as Adam to learning the objectives? Will this have any impact on the loss function $\mathcal{L}_{NCE}(\boldsymbol{\varPhi})$ and the reguarlizer $\mathcal{R}(\boldsymbol{\varPhi},\boldsymbol{L})$?

4). It seems that the concrete form of the lower-bound $\alpha\lambda C(X)$ is not described in the manuscript. I checked the mathematical proof in the appendix, and found that this lower-bound contains the learning parameter $\boldsymbol{L}$ and $\boldsymbol{\varPhi}$. In this case, can we still say that the lower-bound $C(X)$ only depends on the training data $X$?



**Limitations:**

I think there is no potentially negative social impact of this paper. For improvement suggestions, please see the above strengths-weaknesses and questions.

**Strengths And Weaknesses:**

Overall, this paper is interesting and easy to follow. I think it is worth to share this paper to the machine learning community, and the results seem to have the potentiality to emphasize the importance of learning low-dimensional contrastive embedding . Here are some pros and cons of this paper:

**Strengths:**

1). This paper is well-written. The storyline is very smooth to me. Fig. 1, Fig. 2, and Fig. 3 clearly depict the idea, motivation, and effectiveness of the proposed method. It has a strong motivation, which considers a generic case of contrastive learning. Especially, section 3 starts with an interesting and convincing example which illustrates that the hypercube corner number is significantly larger than the data point number.

2). The issue of high-dimensional contrastive embedding pointed out in this paper is important. And I think this contribution is even more valuable than the proposed method itself. Nowadays, it is actually a common issue that the contrastive embedding has very high dimensionality which may lead to the curse of dimensionality. However, as the authors reviewed in the introduction, when we simply reduce the dimensionality of the output layer, it will yet cause the insufficient instance discrimination and harm the generalizability. So this phenomenon indeed leads to an awkward dilemma for us.

3). The proposed idea is non-trivial and concise, and it is well supported by the theoretical analyses. This paper proposes using the additional sparse-regularized layer to reconstruct the high-dimensional features, which I think can well solve the dilemma I mentioned above. More importantly, this idea is supported by the theoretical analyses in Theorem 1, 2, and 3, where the lower-bound of the distance contrast is explicitly given so that the similarity and dissimilarity between pairwise instances can be distinguished from each other.

4). The proposed method is built and implemented in a rather generic way. The authors conduct extensive experiments on multiple domains of data, and the improvements are also non-trivial. I find that the authors further supply the comparison experiments with the distillation-based methods. In the results, the proposed method can consistently outperform the distillation-based methods, and it seems that this can also demonstrate that why we need the new reconstruction layer for solving the issue of high-dimensional contrastive embedding.

**Weaknesses:**

1). As Theorem 1 is an existing work, the author should introduce the related preliminary of Eq. (5) more carefully. For example, does the convergence condition $\lim_{H\rightarrow\infty} var[D_{\boldsymbol{\varPhi}}(x, x_i)/E(D_{\boldsymbol{\varPhi}}(x, x_i))]$ have any relationships with the final conclusion of the distance contrast? This is not very clear to me.

2). It seems that the authors only conduct experiments on the regular positive-negative CL baselines. Does the proposed method work well with the negative-free CL approaches? As we know, currently the negative-free CL has also show encouraging performance which is even better than the positive-negative CL in some cases.

**In summary:**

By considering the above pros and cons, I would believe this is a strong paper in both theories and experiments. I think the quality of paper undoubtedly reaches the acceptance bar of NeurIPS, so I would like to vote for a “clearly accept” and give a score 8 for it. In the rebuttal, I want the authors to further clarify and solve my concerns that I mentioned in the weaknesses and questions.

---

> ### Author Response · Authors · 2022-08-02
> **Response to Reviewer eAPH**
>
> Thank you for your positive and constructive comments! Our point-to-point responses are provided below.
>
> ---
>
> **Comment_1:** What is the relationship between $\lim_{H\rightarrow\infty} var[D_{\boldsymbol{\varPhi}}(x, x_i)/E(D_{\boldsymbol{\varPhi}}(x, x_i))]$ and the conclusion Eq. (5) in Theorem 1?
>
> **Response_1:** Here the convergence condition $\lim_{H\rightarrow\infty} var[D_{\boldsymbol{\varPhi}}(x, x_i)/E(D_{\boldsymbol{\varPhi}}(x, x_i))]$ is a sufficient (but not necessary) premise to ensure that the distance contrast in Eq. (5) will approach to 0. According to the existing work [Beyer et al., ICDT’99], such a convergence condition will be satisfied when $\boldsymbol{\varPhi}(x)$, $\boldsymbol{\varPhi}(x_1)$, $\boldsymbol{\varPhi}(x_2)$, and $\boldsymbol{\varPhi}(x_n)$ are  i.i.d. Notably, the above i.i.d. property [Dasgupta et al., NeurIPS’20] is actually based on the well-known assumption that the original data points $x$, $x_1$, $x_2$, $x_n$ are i.i.d. (as we discussed in the Line 156-166 in our manuscript).
>
> ---
>
> **Comment_2:** Adding more experiments on negative-free CL.
>
> **Response_2:** Although we implement our method on CL models that use both positive and negative samples, our proposed CLLR can also work with negative-free models. We follow the reviewer’s suggestion to conduct experiments on negative-free CL baselines (BYOL [Grill et al., NeurIPS’20] and SimSiam [Chen et al., CVPR’21], merely using positive pairs) to validate the effectiveness of our proposed method. As shown in the following table, our method can consistently improve the compared methods upon themselves (TOP1 and TOP5 accuracy on *ImageNet-100* with 500 training epochs and batch_size = 1024/4096). We have added the additional results to the revised appendix.
> | Method | #1024_TOP1 | #1024_TOP5 | #4096_TOP1 | #4096_TOP5 |
> | :--- | :----: | :----: | :----: | :----: |
> | BYOL | 61.3 | 91.8 | 74.9 | 91.9 |
> | SimSiam | 70.9 | 91.9 | 73.6 | 92.8 |
> | CLLR+BYOL | 63.1 | 92.7 | **76.5** | 93.0 |
> | CLLR+SimSiam | **72.2** | **92.9** | 75.8 | **93.8** |
>
> ------
>
> **Comment_3:** The finally used dimensionality for fine-tuning.
>
> **Response_3:** As the reviewer considered, we use the low-dimensional (256-dimensional) features in our fine-tuning classification experiments. We have also added the explanation to the revised paper, clarifying the feature dimensionality in experiment settings.
>
> ---
>
> **Comment_4:** The holding of Eq. (3).
>
> **Response_4:** Here we follow the reviewer’s suggestion to provide more detailed calculations for Eq. (3). Specifically, according to the definition of gamma function, we have that
>
> $
> \lim_{H\rightarrow\infty}(\pi^{H/2}/(H\cdot\Gamma(H/2)))/2^{H-1}=\lim_{H\rightarrow\infty}(\pi^{H/2}/(H\cdot\int_{0}^{\infty}t^{H/2-1}\text{e}^{-t}dt))/2^{H-1}\leq\lim_{H\rightarrow\infty}(\pi^{H/2}/(H\cdot\int_{1}^{2}t^{H/2-1}\text{e}^{-t}dt))/2^{H-1}.
> $
>
> By further using the *mean-value theorem*, we have
>
> $
> \lim_{H\rightarrow\infty}(\pi^{H/2}/(H\cdot\int_{1}^{2}t^{H/2-1}\text{e}^{-t}dt))/2^{H-1}\leq\lim_{H\rightarrow\infty}(\pi^{H/2}/(H\cdot\text{e}^{-2}))/2^{H-1}\leq\lim_{H\rightarrow\infty}\pi^{(H-1)/2}/2^{H-1}.
> $
>
> Finally, it is easy to obtain that $\lim_{H\rightarrow\infty}\pi^{(H-1)/2}/2^{H-1}=\lim_{H\rightarrow\infty}(\pi/4)^{(H-1)/2}=0$, which is Eq. (3) in our manuscript. We have added detailed explanations in the appendix to clarify this point.
>
> ---
>
> **Comment_5:** Using other optimizers for model training.
>
> **Response_5:** Since our proposed reconstruction loss and regularizer are differentiable almost everywhere, we can employ some other optimizers such as Adam to minimize the learning objective of our CLLR. Specifically, here use the Adam optimizer to train our model on *CIFAR-10* dataset (batch_size = 256), and we record the corresponding training/test errors (%) after 100, 200, and 400 epochs. In the following table, we observe that both SGD (learning_rate = 5e-3) and Adam can converge well after 400 epochs. Therefore, our proposed method has good compatibility with existing (stochastic) optimizers. We have added the additional results to the revised appendix.
> | Optimizer &nbsp; | &nbsp; #100-eps | &nbsp; #200-eps | &nbsp; #300-eps | &nbsp; #400-eps |
> | :--- | :----: | :----: | :----: | :----: |
> | SGD | 30.2±5.3/35.8±4.3 | 10.8±2.1/15.8±2.3 | 3.3±1.8/10.4±2.3 | 2.1±1.1/6.9±1.3 |
> | Adam | 20.2±4.3/25.4±3.3 | 14.1±1.9/18.8±4.1 | 3.4±1.5/10.5±3.4 | 2.4±1.2/7.2±2.1 |
>
> ---
>
> **Comment_6:** Does the error bound merely depend on the training data $\mathcal{X}$ ?
>
> **Response_6:** Yes, we agree with the reviewer that the formulation of the error bound in Eq. (12) contains the feature embedding $\boldsymbol{\varPhi}$ and the parameters of the sparse projection layer. However, it is worth pointing out that such feature embedding and the sparse projection layer are learned from the training data. Therefore, the error bound is still mainly influenced by the training data $\mathcal{X}$ (as well as $\alpha$ and $\lambda$).

---

> > ### Comment · Reviewer_eAPH · 2022-08-08
> > **Thanks for the authors' reply!**
> >
> > Thanks for the detailed derivation and additional experiments. I found the response convincing, and my concerns are well addressed. I still think this should be a good paper for the community.
> >
> > If possible, please also include the new explanations and experiments in the final manuscript.

---

### Official Review · Reviewer_cX3t · 2022-07-12

**Rating:** 3
**Confidence:** 4
**Soundness:** 3 good
**Presentation:** 3 good
**Contribution:** 2 fair

**Summary:**

This paper discusses the contrastive embedding in low-dimensional space. The authors mentioned that contrasitive embedding in high dimensional space would result in the curse of dimensionality and making it difficult to capture  the underlying similarity between pairwise instances. Then the authors propose a new formulation of contrastive embedding under low low-dimensional space, and provide the corresponding convergence analysis and lower bound of min-max distance ratio. The experimental results were provided to show the advantages of the proposed algorithm.

**Questions:**

It would be great if the author could address the questions in weakness and comments.

**Strengths And Weaknesses:**

Strength:

1. The paper has a good motivation, that is contrasitive embedding in high dimensional space would result in the curse of dimensionality and making it difficult to capture  the underlying similarity between pairwise instances. Based on the motivation, the authors proposed a new algorithm and give the corresponding theoretical analysis.
2. The presentation of the paper is good. Have a clear problem setting, and provide a clear description of the target algorithm. For the theoretical analysis, the authors provide a clean derivation.



Weakness:

1. The authors claims that contrasitive embedding in high dimensional space would result in the curse of dimensionality and making it difficult to capture  the underlying similarity between pairwise instances. Thus, in the experiments of ablation study, the extreme cases should be provided to show that the contrasitive embedding in high dimensional space could conduct very poor performance. That is not included in Table 1.

2. It is good to provide section 2.1 to show the motivation of this paper. If the example could be more direct to show the disaster of contrasitive embedding in high dimensional space, that would improve the quality of the paper.

3. It is fine for this paper to plug the low-dimensional mapping into contrastive embedding. However, the novelty and contribution is not obvious to reach the bar of Neurips.



===============================

Thanks the authors for providing the more experimental results. However, I do not think the new results convince me. I would like to decrease my score to 3.

---

> ### Author Response · Authors · 2022-08-02
> **Response to Reviewer cX3t**
>
> We are grateful for your valuable comments! We have carefully addressed each of the concerns below.
>
> ---
>
> **Comment_1:** High-dimensional failed cases should be added to the ablation study.
>
> **Response_1:** We agree with the reviewer that the high-dimensional cases are important baselines. Actually, Tab. 1 in our manuscript already contains a case of high-dimensional contrastive embeddings (*i.e.*, the 2048-dim. w/o $\mathcal{R} (\boldsymbol{\varPhi}, \boldsymbol{L})$), which is a popular setting in existing CL methods. Here we also follow the reviewer’s suggestion to further add two more high-dimensional (3072 and 4096 dimensional) contrastive embeddings into our ablation study as additional baselines for comparison. In the following table, we can clearly observe that the classification accuracy is not really improved (or even be worse) when the feature dimensionalities are higher than 2048, and we think this is because data points are sparsely distributed in the very high-dimensional space and the pairwise similarities can hardly be captured. By using our proposed CLLR, we can learn a low-dimensional contrastive embedding to avoid such an issue, and thus successfully improving the classification accuracy. We have also added the new baselines to our revised paper.
> | Method | &nbsp; STL-10_#100-eps &nbsp; |&nbsp; STL-10_#400-eps &nbsp; | &nbsp;  CIFAR-10_#100-eps &nbsp; | &nbsp;  CIFAR-10_#400-eps &nbsp; |
> | :--- | :----: | :----: | :----: | :----: |
> | 4096-dim. w/o $\mathcal{R} (\boldsymbol{\varPhi}, \boldsymbol{L})$ | 55.1±1.1 | 75.2±3.1 | 65.1±1.9 | 85.4±4.2 |
> | 3072-dim. w/o $\mathcal{R} (\boldsymbol{\varPhi}, \boldsymbol{L})$ | 54.4±3.1 | 75.2±2.1 | 67.2±3.5 | 86.9±6.1 |
> | 2048-dim. w/o $\mathcal{R} (\boldsymbol{\varPhi}, \boldsymbol{L})$ | 56.3±2.1 | 76.2±1.1 | 66.3±3.1 | 89.3±2.1 |
> | 256-dim. w/o $\mathcal{R} (\boldsymbol{\varPhi}, \boldsymbol{L})$ | 55.3±4.1 | 74.2±2.1 | 64.3±5.1 | 88.3±3.1 |
> | 256-dim. w/ nuclear (ours) | 56.3±3.2 | &nbsp; &nbsp; **79.2±0.2** ✔ | 65.2±5.5 | &nbsp; &nbsp; **93.1±1.3** ✔ |
>
> ---
>
> **Comment_2:** Examples should be given in Section 2.1 to show the disaster of contrastive embedding in high-dimensional space.
>
> **Response_2:** Thanks for the suggestion! We want to clarify that we had provided an intuitive example in Section 2.1 to demonstrate the disaster/unreasonableness of the high-dimensional contrastive embedding. Specifically, Fig. 2 in our manuscript shows three distance distributions (namely the pairwise similarity between each two data points) of the 256, 512, and 2048 dimensional contrastive embeddings on *STL-10* dataset. We can observe that the (2048)high-dimensional embedding tends to predict almost the same distances for each pair of data points, *i.e.*, the sharp distribution in Fig. 2(c). However, such a phenomenon implies that the learned high-dimensional contrastive embedding fails to predict small distances for intra-cluster data pairs and large distances for inter-cluster data pairs, respectively, so that the discriminating features can hardly be captured by the learning algorithm. This inspires us to propose a new algorithm to learn a low-dimensional contrastive embedding.
>
> ---
>
> **Comment_3:** Novelty/contribution is not obvious.
>
> **Response_3:** We would like to kindly highlight the novelty and contribution of our paper. Specifically, this paper aims to address the unreliable similarity prediction caused by the high dimensionality in existing CL methods. Although simply reducing the dimensionality of output layers and distillation methods can be employed to obtain the low-dimensional embedding, they cannot encourage instance discrimination or render reliable similarity predictions, so the curse of dimensionality remains in CL. To this end, we propose a novel sparse projection layer to adaptively learn a low-dimensional latent feature embedding. The following table summarizes the novelty/motivation of our paper.
> | Method | &nbsp; #Instance Discrimination &nbsp; | &nbsp; #Reliable Similarity Prediction &nbsp; |
> | :--- | :----: | :----: |
> | Reducing Dim. of Output Layers | ✘ | ✔ |
> | Distillation-Based Methods | ✔ | ✘ |
> | Our Proposed CLLR | **✔** | **✔** |
>
> The proposed method has a completely new architecture (*i.e.*, the sparse projection) and a new formulation (*i.e.*, the low-dimensional regularization) which have not been explored before. To the best of our knowledge, we are the first to propose learning contrastive embedding in low-dimensional space. The novelty of our paper is also confirmed by the other three reviewers. We provide both theoretical analyses and multi-domain experiments to validate the effectiveness of our method. Therefore, our paper has a potential impact to emphasize the significance/necessity of learning low-dimensional contrastive embedding (as *Reviewer VTWD* and *Reviewer eAPH* acknowledged). We would be grateful if the reviewer could provide more suggestions for further clarifying the novelty or contribution of our paper.

---

> > ### Comment · Reviewer_cX3t · 2022-08-09
> > **thanks for answering my questions**
> >
> > Thanks for answering my questions. For the question 1, I did not find the very poor performance in the high-dimensional space. I still believe it is important to show the extreme case to verify your motivation. I keep my score.

---

> > > ### Author Response · Authors · 2022-08-09
> > > **Thank you the the reply**
> > >
> > > Dear Reviewer cX3t,
> > >
> > > Thank you for the reply!
> > >
> > > As the reviewer's new comment is posted ten minutes before the end of the Author-Reviewer discussion, please allow us to answer the question in the current Reviewer-Area Chair discussion period.
> > >
> > > >For the question 1, I did not find the very poor performance in the high-dimensional space.
> > >
> > > Please note that here the term "poor performance" should be a comparative concept/result.
> > >
> > > In Tab. 1, the accuracy rates of the high-dimensional cases on *STL-10* (75.2) and *CIFAR-10* (85.4) datasets are significantly lower than the accuracy rates of our low-dimensional cases (79.2 and 93.1, respectively). Such significance is strictly validated by the *t*-significance test. Please also note that we should mainly compare the accuracy rates in the 3rd and 5th columns (namely 400 epochs, not 100 epochs), where the models are sufficiently trained.
> > >
> > > Therefore, there is still room in the baseline method for improving the final performance, and we believe this clearly validates our motivation.
> > >
> > > Looking forward to your further reply!
> > >
> > > Best regards,
> > >
> > > Authors of Paper ID 5171

---

> > > ### Comment · Reviewer_eAPH · 2022-08-10
> > > **I feel that the authors have actually addressed your Question (Comment) 1.**
> > >
> > > Dear Reviewer-cX3t,
> > >
> > > I feel that the authors have actually addressed your Question (Comment) 1.
> > >
> > > For one thing, I think the extremely bad cases (namely, very poor performance) may not really exist because the well-known CL methods (such as SimCLR, CMC, and CO2) could already get good and acceptable classification accuracy. In this case, we may not expect some good baselines to produce very awful accuracy rates, so it seems unrealistic and unfair to ask authors to report some extremely bad accuracy rates for their compared baseline method. Meanwhile, I also observe that the accuracy rates indeed begin to be worse when the dimension is larger than 2048, so I believe this result should already illustrate the failed cases of high-dimensional settings in terms of accuracy rate.
> > >
> > > For another, I find the authors' practice in Tab. 1 very useful and technically sound (also mentioned by Reviewer-VTWD), where they inversely validate that the existing results of high-dimensional settings can still be vastly improved. This follows a mainstream paradigm of ablation study, and it implies that the high-dimensional cases of existing CL may not be that great (i.e., “poor performance” in your words). Thus I think such indirect practice also effectively verifies the motivation and it seems more friendly than straight reporting the extremely bad performance.
> > >
> > > Furthermore, I agree with the authors that Fig. 2 provides another angle (namely the similarity/distance prediction) to reveal the unreasonableness of the high-dimensional setting. In Fig. 2(c), all pairwise distances tend improperly to be the same, and it is clear that the learned representation is not our desideratum (inter-class and intra-class distances should be different). So this phenomenon further shows an unreasonable/poor result of the high-dimensional baseline.
> > >
> > > All in all, I do think the failed cases of the high-dimensional settings are well demonstrated by the authors. I totally agree with you that showing failed cases is important to verify the motivation, and I also believe the authors have indeed done great efforts on this point. In general, I like this paper and I have given it the highest score of all my reviewed CL-related papers, so I want to keep my recommendation to support it.
> > >
> > > Thank you for reading! Discussions are welcome :-)
> > >
> > > Best,
> > >
> > > Reviewer-eAPH

---

### Author Response · Authors · 2022-08-06
**Dear Reviewers**

Dear Reviewers:

Thanks a lot for your time and efforts!

We are writing to you to express our concerns about the rebuttal process.

Do you have any new questions after reading our responses? We are glad to provide targeted answers if you have more concerns.

Kind regards,

Authors of Paper ID 5171

---

### Meta-Review · Area_Chair_hnUZ · 2022-08-26

**Recommendation:** Accept
**Confidence:** Less certain

**Metareview:**

This paper proposes a new  plug-in style method for contrastive self-supervised learning to yield low-dimensional embeddling. The authors argue that high-dimensional embeeding induced by negative pair-based constrastive learning leads to "curse of dimensionality", which might harm the generalization performance. So, the authors employs a sparse project layer on the top of typical CSSL encoder for low-dimensional representation.

The scores of the reviewers are split: two strong acceptance (8 and 7) and two rejection (3 and 4).
Unfortunately, the overall situation does not changes (8, 6->7, 4->3, 3->4) after rebuttal and two discussion periods.

AC carefully read the paper, the author rebuttal, and discussion comments.

All reviews agree that this paper has novelty, interesting idea, theoretical justification, and extensive-promising results.
The main controversal point is the motivation that "curse of dimensionality" by high-dimensional representation harms the generalization performance. On this issue, three reviewers debated very enthusiastically. It is difficult to decide whether the performance degradation of high-dimensional embedding is meaningful or not. Also, it is not very clear that this degradation is directly caused by the curse of dimensionality. However, empirical results show the proposed low-dimensional method consistently improves the high-dimensional method with sparse regularization including negarive-free CL methods. Practically, low-dimensional embedding has advantages in memory footprint and inference time in the real-world applications, which these merits are not discussed.

Overall, this paper can contrubte to ML community despite its controversial motivation. So AC recommedns accepting this paer.
Please clarify the motivation considering two reviewers' comments.



**Award:**

No

---

### Decision · Program_Chairs · 2022-09-14

Accept